# Clustering Arid Rangelands Based on NDVI Annual Patterns and Their Persistence

**Ernesto Sanz** [1,2,*] , **Juan José Martín Sotoca** [1,2] , **Antonio Saa-Requejo** [1,3] , **Carlos H. Díaz-Ambrona** [1,4] , **Margarita Ruiz-Ramos** [1,4] , **Alfredo Rodríguez** [1,5] and **Ana M. Tarquis** [1,2]

1    CEIGRAM, Universidad Politécnica de Madrid, 28040 Madrid, Spain
2    Grupo de Sistemas Complejos, Universidad Politécnica de Madrid, 28040 Madrid, Spain
3    Evaluación de Recursos Naturales, ETSI Agronómica, Alimentaria y Biosistemas, Universidad Politécnica de Madrid, 28040 Madrid, Spain
4    AgSystems, ETSI Agronómica, Alimentaria y Biosistemas, Universidad Politécnica de Madrid, 28040 Madrid, Spain
5    Departamento de Análisis Económico y Finanzas, Universidad de Castilla-La Mancha, 45071 Toledo, Spain
*    Correspondence: ernesto.sanz@upm.es

**Abstract:** Rangeland ecosystems comprise more than a third of the global land surface, sustaining essential ecosystem services and livelihoods. In Spain, Southeast Spain includes some of the driest regions; accordingly, rangelands from Murcia and Almeria provinces were selected for this study. We used time series metrics and the Hurst Exponent from rescale range and detrended fluctuation analysis to cluster different rangeland dynamics to classify temporally and spatially diverse rangelands. The metrics were only calculated for three time periods that showed significant NDVI changes: March to April, April to July, and September to December. Detrended fluctuation analysis was not previously employed to cluster vegetation. This study used it to improve rangeland classification. K-means and unsupervised random forest were used to cluster the pixels using time series metrics and Hurst exponents. The best clustering results were obtained when unsupervised random forest was used with the Hurst exponent calculated with detrended fluctuation analysis. We used the Silhouette Index to evaluate the clustering results and a spatial comparison with topographical data. Our results show that adding the Hurst exponent, calculated with detrended fluctuation analysis, provided a better classification when clustering NDVI time series, while classifications without the Hurst exponent or with the Hurst exponent calculated with the rescale range method showed lower silhouette values. Overall, this shows the importance of using detrending when calculating the Hurst exponent on vegetation time series, and its usefulness in studying rangeland dynamics for management and research.

**Keywords:** NDVI; multiscaling; vegetation dynamics; rangelands; detrended fluctuation analysis; random forest

## 1. Introduction

Ecosystems were considered complex systems with non-linear dynamics in space and time for more than three decades [1–4]. However, only recent research focuses on tackling the complexity of ecosystem temporal dynamics with various methodologies [5–12]. As an eco-social system, rangelands comprise 30–40% of the Earth's landmass, supporting approximately 1 billion people [13,14]; this makes them suitable land types to study ecosystem dynamics with significant human activity effects. This type of land is heavily affected by land degradation, affecting 73% of all rangelands [15–19]. Land degradation reduces biological productivity, ecosystem functions, and complexity [20,21].

Climate change and social-economic trends are some of the main challenges in rangeland conservation, often with interactive and synergy responses [22,23]. An integrated

approach to land management is required to address these issues. Understanding the dynamics and characteristics of rangelands is a vital part of their conservation [4,24–26]. The Normalized Differentiated Vegetation Index (NDVI) was widely used to monitor, assess, and classify vegetation [27–31]. Moreover, more recently, supervised and unsupervised machine learning was used to classify rangeland pixels based on values of NDVI at the spatial level. Summary statistics of NDVI time series were also used to study spatiotemporal data [32–35]. The unsupervised classification does not require labelled data. K-means and ISODATA algorithms are commonly used in unsupervised land cover and crop classification. However, these algorithms are susceptible to outliers, high dimensionality, and noise. Unsupervised Random Forest (URF) was previously used with other biological data, such as genomic sequence data [36] and vegetation [37]. Different metrics to measure the fitness of the cluster were developed. The Silhouette Index is an excellent internal cluster validation metric, more robust than other metrics, such as the Rand Index and Dun Index [38–41].

A method to study the complexity of time series is their fractal character using the Hurst exponent [42]. This method was developed to measure the persistence (H > 0.5) or antipersistence (H < 0.5) of a time series. This analysis can be calculated using the Rescaled Range (R/S) method, named the Hurst Index (HI, [42]). Another method uses detrended fluctuation analyses (DFA), which removes tendencies of the time series before calculating the Hurst exponent (H2), the generalised Hurst exponent for q = 2 [43]. Both methods were used in long-term ecosystem dynamics on vegetation [8,10,44,45]. Another application was to localise changes in those dynamics, such as those affected by fire [46]. When a time series is persistent, the trend of that time series will continue in the same direction. However, if a time series is antipersistent, the trend will be followed by the opposite (e.g., if the trend were increasing, it would be followed by a descent). If the Hurst exponent is close to 0.5, the time series will follow a random process, such as a random walk.

The Hurst exponent was applied to the NDVI time series to quantify the long-term memory as well as their trend. Long-term memory is affected by land use, land changes, and climate change, making it useful for rangeland managers. Topographical variables were also linked to Hurst exponent values [5,6]. Several authors used it to map rangelands or other vegetation, and comment on their connection with slope and elevation values [5,8,10,47–50]. However, to our knowledge, integrating the system persistence characteristics with the NDVI annual pattern to classify rangelands was not yet accomplished. The Hurst exponent represents vegetation dynamics and NDVI time-series summary statistics represent the vegetation types. This research aims to provide new insights into a spatially complex eco-social system, where aridity, land degradation, and climate change restrict agricultural practices and ecosystem services [51,52]. Clustering rangeland pixels in arid areas can be used to prioritise field visits where different vegetation dynamics and trends are found.

The present study attempted rangeland classification, including the Hurst exponent. Two Hurst exponent methods (HI and H2) were used to evaluate the influence of the DFA in capturing vegetation dynamics. Additionally, two different machine learning methods (k-means and URF) were applied to decide which provided a more accurate outcome based on the Silhouette Index.

## 2. Materials and Methods

### 2.1. Area of Study

Three agricultural regions of Southeast Spain were selected (Figure 1): Los Velez in the province of Almeria, and the Northwest and Northeast in the province of Murcia, which will be called Murcia-NW and Murcia-NE, respectively, for clarity. These three regions have a Mediterranean arid climate with an average annual precipitation of less than 300 mm, although with regional variations [53]. The spatial resolution used was 250 m/pixel. This spatial resolution matches the resolution used by most stakeholders in the Spanish agricultural insurance system. The pixel selection was provided by ENtidad Estatal de Seguros Agrarios (ENESA, Ministerio de Agricultura, Pesca y Alimentación,

Government of Spain), using the Sistema de Información Geográfico de Parcelas Agrícolas (SIGPAC [54]) and the Mapa Forestal Español (MFE, Spanish Forest Map). Firstly, pixels categorised as rangeland were selected using the SIGPAC. Secondly, using the previous selection, pixels with a tree coverage higher than 15% were discarded to ensure a low tree coverage, based on the MFE. Three thousand six hundred and fifty-four (3654) pixels of rangelands were selected, consisting of grasslands, shrublands, and open woodlands.

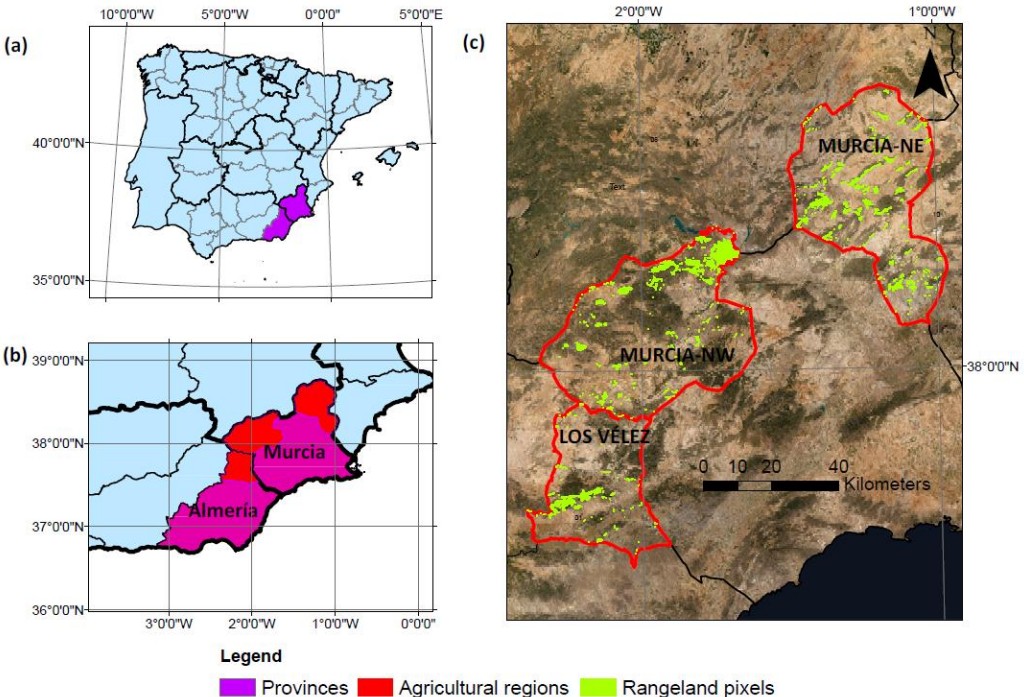

**Figure 1.** Location of the study area. (**a**) Selected Provinces. (**b**) Selected agricultural regions of Almeria and Murcia. (**c**) Selected pixels in three agricultural regions of Almería and Murcia. Source basemap: Invierno 2020. Gobierno de España y Comunidad Autónoma de Murcia. CC-BY 4.0 scne.es 2020.

The three regions are mainly located in mountainous areas. The Murcia-NE region is mainly a mix of grassland and shrubland; Murcia-NW is dominated by sparse woodland mixed with shrubs; and in Los Velez, grasslands and shrublands are the primary vegetation with minimal areas of sparse woodland. These regions include areas with different aspects and changing slopes and elevations.

*2.2. Data Collection*

2.2.1. NDVI Data

The NDVI data were collected from MOD09A1.006, using the AppEEARS tool [55], and downloading the RED (band 1) and NIR (band 2) values for the target areas. This tool has a 250 m spatial resolution, collecting a set of 3654 pixels and an 8-day temporal resolution from the beginning of 2000 to 2019, a total of 20 years of data. R software [56] was used for each pixel series to calculate the NDVI, using Equation (1) below. The 8-day temporal resolution was transformed to a 10-day resolution as used by the Spanish indexed agricultural insurance.

$$\text{NDVI} = 100 \times \frac{\text{NIR} - \text{RED}}{\text{NIR} + \text{RED}} \tag{1}$$

The possible NDVI values range from 0 to 100. The obtained NDVI values were then checked for quality. The data were deleted if they were not categorised as ideal quality (quality values in band from AppEEARS, less than 0.01%). The gaps were filled using running averages with a gap interval of seven dates. The time series were then smoothed

using the Savitzky–Golay method [57], with a window size of 9 selected, based on the best-fitted outputs.

### 2.2.2. GIS Data

A Digital Elevation Model (DEM, 10 m resolution) was downloaded from the Copernicus website [58], and ArcGIS software v. 10.8.1 [59] was used to calculate the slope based on the DEM. These two datasets, and the variables used in the clustering analysis (Hurst exponent and NDVI summary statistics), were used to compare the clustering results through boxplots for a visual comparison.

### 2.3. Fractal Analysis

### 2.3.1. Rescale Ranged Hurst Exponent

Hurst Index analysis was used to analyse the persistence of NDVI in each area [42]. For this index, the package "pracma" (version 1.9.9) [60] was used in R Software. This index splits the time series into $\tau$ subseries. Each subseries calculates the mean and cumulative sum of the mean to calculate the range ($R(\tau)$). This range is divided by each subseries standard deviation ($S(\tau)$). The Hurst exponent (HI) is then calculated using the following formula and by averaging each subseries, where c is a constant of proportionality, $\tau$ is the time span, and H is the Hurst scaling exponent.

$$\frac{R(\tau)}{S(\tau)} = c\tau^{H}$$

(2)

### 2.3.2. Multifractal Detrended Fluctuation Analysis

A Mann–Kendall test [61,62] was applied to the whole temporal series of each pixel. Since most of the NDVI series presented a trend, Multifractal Detrended Fluctuation Analysis (MF-DFA) was used following [43], developed to calculate multifractal properties after removing trends in the time series. The main feature of multifractals is that they are characterised by high variability over wide ranges of temporal or spatial scales associated with intermittent fluctuations and long-range power-law correlations.

The MF-DFA operates on x(i), where $i = 1, 2, \ldots, N$, with $N$ being the series length; $\overline{x}$ represents the mean value, and x(i) are increments of a random walk process around the average $\overline{x}$. The integration of the signal, therefore, provides what is called the 'trajectory' or 'profile':

$$y(i) = \sum_{k=1}^{i} [x(k) - \overline{x}]$$

(3)

Furthermore, the integration will reduce the level of measurement noise present in observational and finite records. Next, the integrated series was divided into $N_s$ = int $(N/s)$, the integer part of non-overlapping segments of equal lengths $s$. The local trend was then calculated for each Ns segment by a least-squares fit, and then the variance was determined:

$$F^2(s, v) = \frac{1}{s} \sum_{i=1}^{s} \{y[(v-1)s + i] - y_v(i)\}^2$$

(4)

for each segment $v$, where $v = 1, \ldots, N_S$. Here, $y_v$(i) is the fitting curve in segment $v$. In this case study, a line was chosen. After detrending the series, the average was performed over all segments to obtain the 2nd-order fluctuation function:

$$F_q(s) = \left\{ \frac{1}{2N_s} \sum \left[ F^2(s, v) \right]^{\frac{q}{2}} \right\}^{\frac{1}{q}}$$

(5)

*H(q)* is the generalised MF-DFA exponent in the function of q. *H(q)* was calculated for the time scales where the fluctuation functions increased linearly to allow detrending

calculations, starting at 32 days. Observing Equations (4) and (5), in the case that q = 2, the equation will be:

$$F_2(s) \propto s^{H(2)} \tag{6}$$

Therefore, H2 = H(2) is the Hurst index estimated using MF-DFA as it was used by [63]. In this study, given that only one exponent (H2) was used, this method will be referred to as DFA.

## 2.4. Variable Selection for Clustering

Summary statistics of the NDVI time series (quartiles 1, 2, 3, and variances) were calculated to analyse vegetation dynamics, similarly to [34,35]. However, the statistics were calculated at different year moments (phases) where NDVI behaves differently across the year. Three periods were chosen when the NDVI experienced more significant changes: Phase 2 (March and the first two ten-day periods of April), Phase 3 (from the last ten-day period of April to the last ten-day period of July), and Phase 5 (September to December) following [64]. For these three periods, the mentioned summary statistics were calculated. The Hurst exponent was then calculated for the whole NDVI time series using two methods, R/S and DFA. Afterwards, clustering techniques were used on the selected summary statistics independently, and with each of the Hurst exponents. The results were compared to topographical data: elevation and slope.

Among all summary statistics and the Hurst exponents, a correlation matrix was applied to select variables that did not have a strong correlation (i.e., <0.75). Principal component analysis was run when strong correlations were present to select the most explanatory variables. Upon selection, clustering analyses were run and compared.

## 2.5. Clustering

Clustering was made using two unsupervised machine learning methods (k-means and URF). The Silhouette Index [39] was used to compare the different classification results and select the best option based on the partition and all proximities for all objects. The Silhouette Index was calculated for clusters A and C, following Equation (7):

$$SI(i) = \frac{b(i) - a(i)}{\max\{a(i), b(i)\}} \tag{7}$$

where a(i) is the average dissimilarity i to all other objects of cluster A, and b(i) is the minimum average dissimilarity of i to the centroid of cluster C.

To study the differences and similarities between the clusters, the adjusted Rand Index was used [65] from the R package "fossil v. 0.4.0" [66], which determines whether two clusters are similar to each other using a contingency table of the two clusters making an all pair-wise comparison.

### 2.5.1. K-Means

K-means was developed by Stuart Lloyd in 1957 and published in 1982 [67]. It is a non-hierarchical technique, and one of the simplest methods to solve clustering problems. James MacQueen first coined this method as k-means in 1967 [68]. This algorithm starts clustering by randomly assigning a K number of centroids. Secondly, it calculates the distance between the data points, and the closest centroid minimises the sum of the square as in Equation (8):

$$d(x, y) = \frac{1}{2} \sum_i (x_i - y_i)^2 \tag{8}$$

The algorithm repeats this process by adjusting the centroids based on the calculated distance, iterating a set number, and converging in a fixed point [69]. In this paper, Hartigan and Wong's method was used [70] with the R package "stats v. 3.6.2" [56]. This method

reassigns point by point, considering the shift in the means after the reassignment of previous points, and it may reassign a point even if it already has an assigned centre.

### 2.5.2. Unsupervised Random Forest

Random forest [71] is a tree-based ensemble method, i.e., methods that generate many classifiers and aggregate their results. It uses bootstrap aggregating (bagging [72]) to calculate a large number of trees based on the fed predictor variables and to select the most voted trees. Random forest is a non-parametric method that builds each tree using a deterministic algorithm based on the three main variables: (1) the number of trees ($nt$); (2) the number of predictors tested on each node ($m$); and (3) the minimal size for each node (*nodesize*). A third of the bootstrap is omitted in each node and is considered out-of-the-bag (OOB) data. These data are used to obtain a classification rate for each node. The variable importance is calculated for the averaged final tree based on the OOB data and their classification rate. Each tree presents a different variable importance, but these are averaged [73]. The R package "randomForest v. 4.6-14" was used [74] to calculate the RF as an unsupervised method, utilising the proximity matrix as predictor variables.

## 3. Results

### 3.1. Variable Selection Approach

All summary statistics between the three selected phases presented a robust linear correlation (>0.75) except for the three variances (Figure 2). Principal component analyses (PCA) were performed with our statistic variables, including the Hurst exponents (HI and H2). Moreover, among those variables with a strong correlation, quartile 3 of phase 5 was chosen for its higher explanatory power (Tables A1 and A2). The three variances and quartile 3 of phase 5 were used separately with HI and H2, and with neither of them.

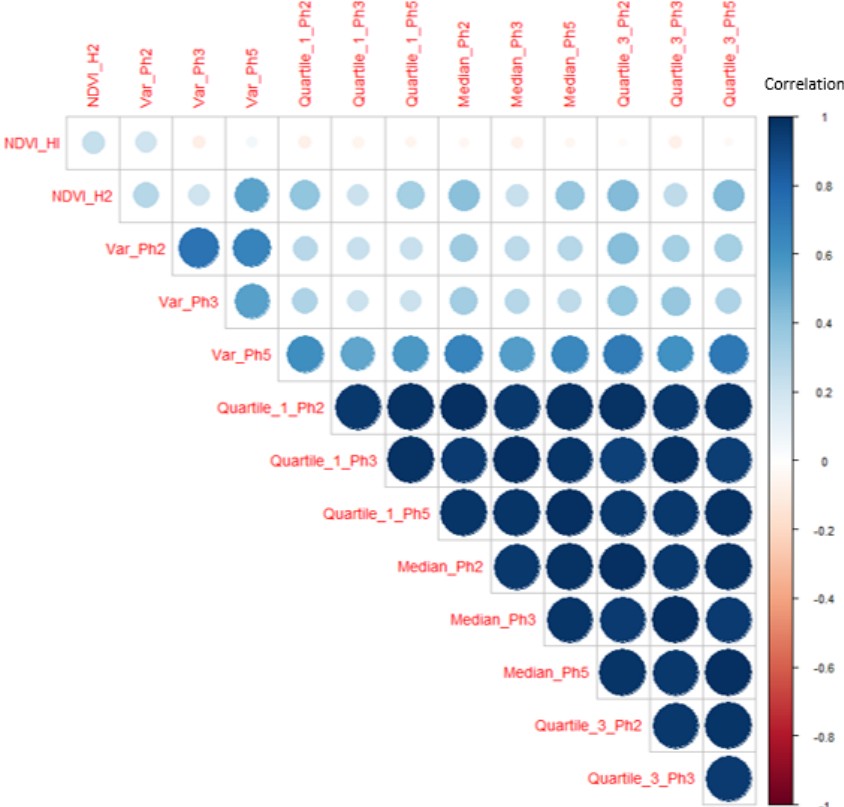

**Figure 2.** Correlation matrix of all variables tested for the three study regions. Large, dark blue circles indicate a high correlation, while small, light blue circles indicate a low correlation. Ph2/3/5 stands for Phase 2/3/5, and Var for variance.

### 3.2. Clustering Analysis

3.2.1. K-Means

The k-mean analyses were applied, using the aforementioned selected variables, for three and four clusters based on the elbow method. The elbow method is a heuristic method to determine the number of clusters in a dataset [75], as shown in Figure 3. K-means clustering was different when three (three-cluster analyses) and four (four-cluster analyses) clusters were used. However, for each cluster number, the results were identical whether no Hurst exponent, H2 or HI were used. The clustering results presented an adjusted Rand Index of 1 among the three-cluster analyses and an adjusted Rand Index of 0.84 when comparing the results of three- and four-cluster analyses. The fourth cluster showed very few pixels with a low Silhouette Index for this cluster, as shown in Figure 4. The Silhouette Index was the same in all k-means analyses with three- and four-cluster analyses (Table 1).

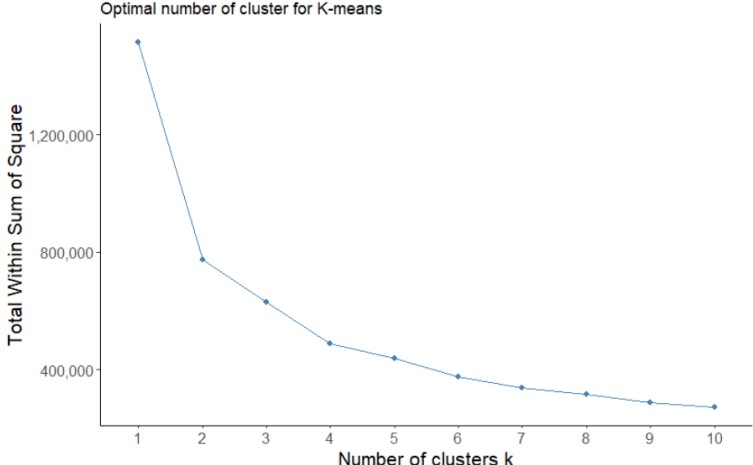

**Figure 3.** Elbow method on the selected variables using k-means clustering. Three- and four-cluster analyses were performed.

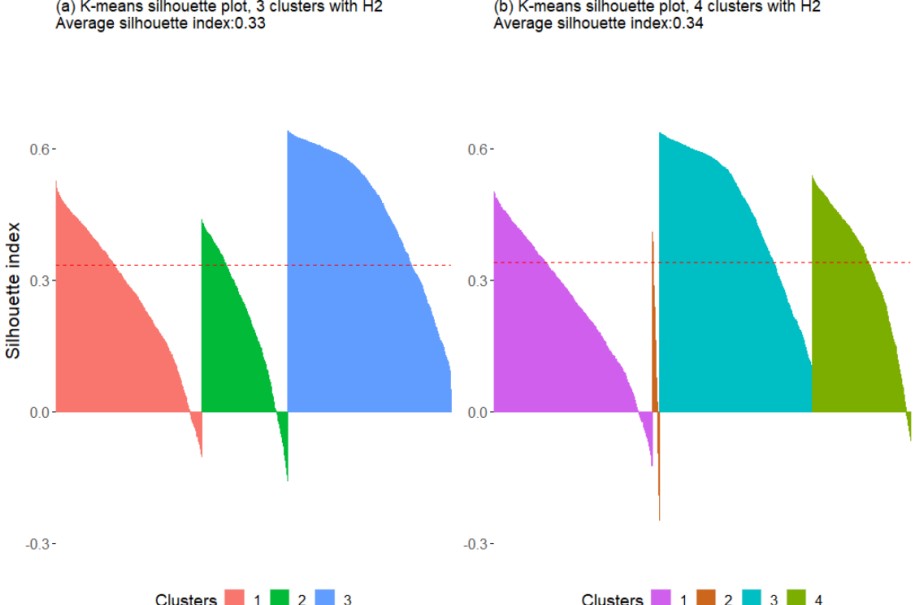

**Figure 4.** Silhouette plots for all pixels using k-means for three (**a**) and four clusters (**b**), showing the Silhouette Index (*y*-axis) for all pixels for each cluster represented on the *x*-axis with H2. The same results were obtained when k-means were run with HI or without HI/H2.

**Table 1.** Average Silhouette Indexes for 3 and 4 clusters for k-means and optimised URF.

| Analysis | K-Means | | Unsupervised Random Forest | |
|---|---|---|---|---|
| | 3 Clusters | 4 Clusters | 3 Clusters | 4 Clusters |
| Without H2/HI | 0.33 | 0.34 | 0.51 | 0.49 |
| With H2 | 0.33 | 0.34 | 0.62 | 0.47 |
| With HI | 0.33 | 0.34 | 0.50 | 0.45 |

3.2.2. Unsupervised Random Forest

Using the elbow method with the partitioning around medoids method showed a similar graphic as using the k-means method, indicating that three and four clusters may be the most appropriate to use (Figure 5).

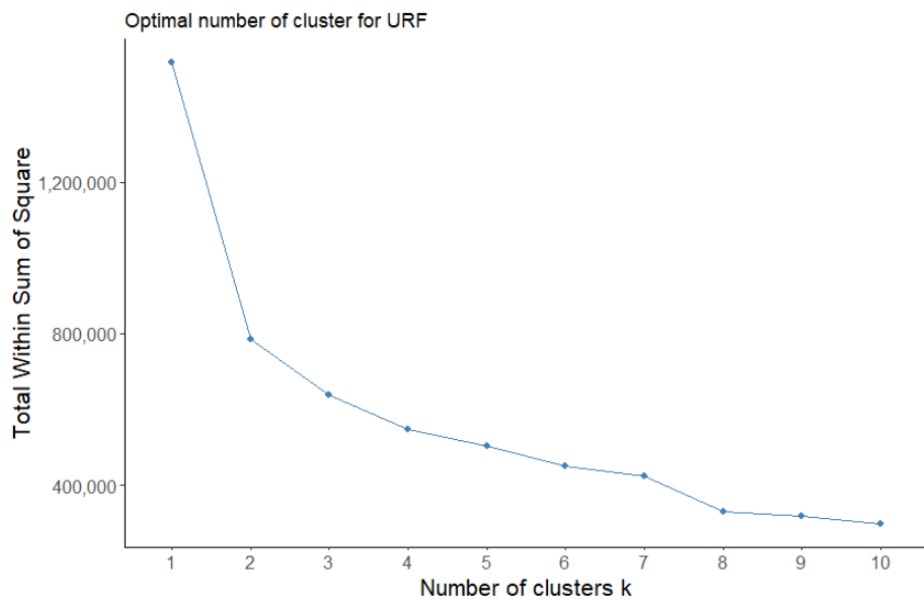

**Figure 5.** Elbow method of the selected variables using partitioning around medoids clustering, where three and four clusters were selected.

The URF has more variables that affect the results: the number of trees (nt) and several variables (m) used for splitting branches. Three or four clusters were used and H2, HI, and no Hurst exponent analyses, were calculated. For each combination, URF was calculated for different nt and m to obtain the analysis with the highest Silhouette Index (Figures A4–A6). Compared with k-means, URF showed higher variability between the results, whether using H2, HI, or no Hurst exponent. The silhouette values from URF were consistently higher when three groups were used for the three analyses regarding the Hurst exponent (Table 1). When four clusters were used in our analyses, the additional fourth cluster showed a low Silhouette Index for that cluster (Figure 6). Therefore, only the URF clustering for three clusters will be discussed with and without the Hurst exponents, focusing on the cluster with the highest Silhouette Index (H2).

The clustering results were more similar between the use of HI and no Hurst exponent than when H2 or HI was used, presenting 0.82 and 0.74 in the adjusted Rand Index, respectively. For all cases, cluster 1 was the most predominant, and cluster 2 had a higher NDVI and variance, while the opposite can be said for cluster 3. These differences were more remarkable when H2 was used. The difference in Hurst exponent (HI or H2, respectively) between the three clusters was more evident when H2 was used. The major differences in clustering among these three analyses were found in cluster 2, that with the highest H2 and NDVI (Figures 7 and 8). These distinct pixels were found mainly in the Murcia-NW region (Figures 9 and A4–A6).

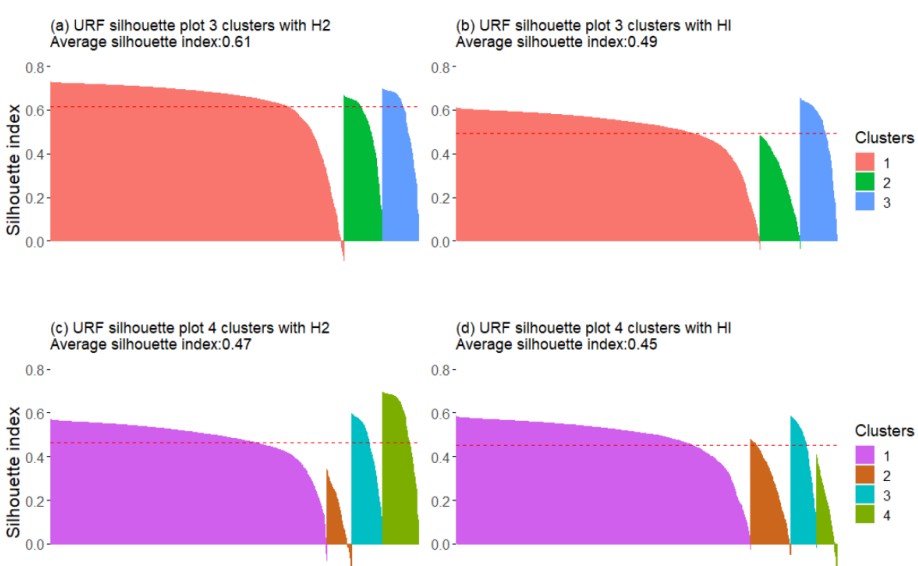

**Figure 6.** Silhouette plots for the different analyses performed with URF. On the left are those performed with H2, from DFA; the analyses performed with HI, from R/S, are on the right. The top graphics are for three clusters and the bottom graphics are for four. Silhouette plots show the Silhouette Index (*y*-axis) for all pixels for each cluster represented on the *x*-axis.

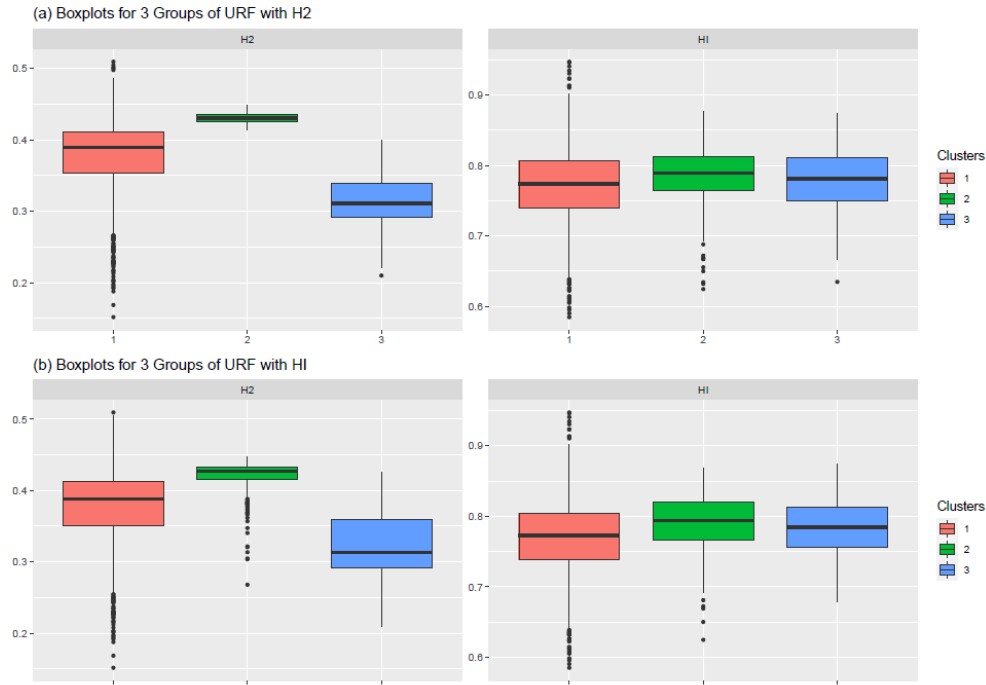

**Figure 7.** Comparison of H2 and HI for all clusters when URF was used with H2 (**top**) and HI (**bottom**) in all study areas.

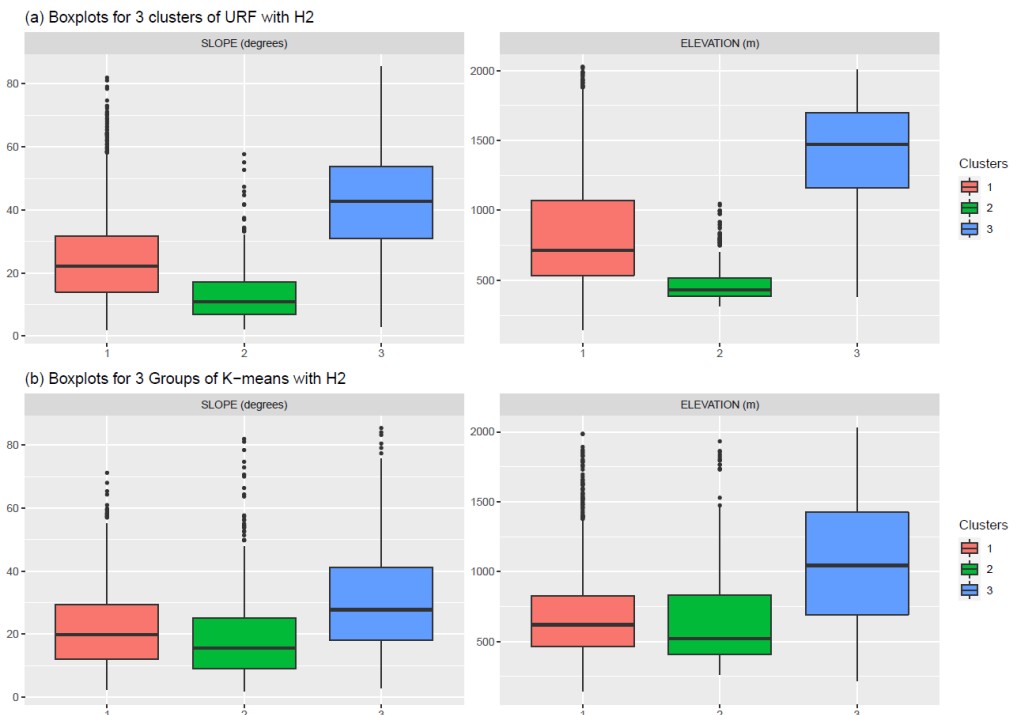

**Figure 8.** Slope and elevation comparison for all clusters when URF (**a**) and k-means (**b**) were used with H2 in all study areas.

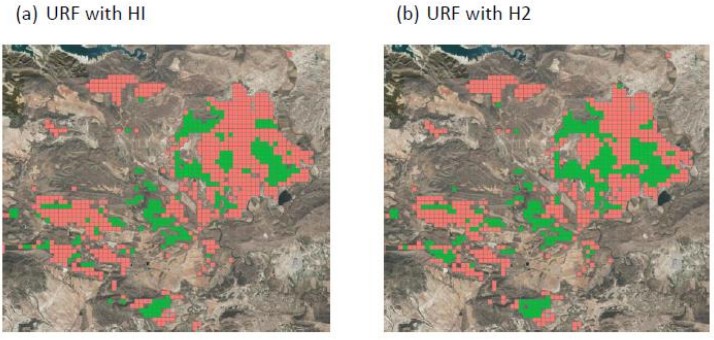

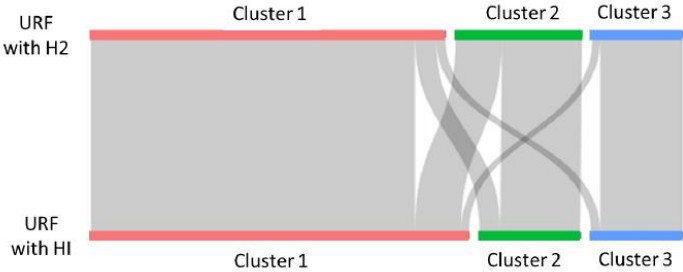

**Figure 9.** (**a**) On the top are the clustering results of URF in the Murcia-NW when HI (**a**) or H2 (**b**) was used, showing cluster 1 and 2 present in this region, while cluster three was not present in this area. (**c**) compares the differences in clustering when HI (**bottom**) and H2 (**top**) were used in URF for all the study areas.

### 3.2.3. Cluster Characterisation

The Hurst exponent from DFA showed a stronger linear correlation with elevation and slope than HI. The same occurred with the selected variables used for the clustering

analyses (Table 2), the variances from phases 5 and 2 (those with the highest correlation to H2 and HI, respectively). These correlations were reflected in the clustering process. When URF with H2 was used, slope and elevation were more heavily differentiated for clusters 2 and 3. These differences were not found when k-means was used since the clustering outcome was the same when H2 substituted HI, or no Hurst exponent was used. Furthermore, slope and elevation showed a more considerable overlap between the clusters on the three-cluster analyses when k-means was used (Figure 8).

**Table 2.** Correlations between H2 and HI with elevation, slope, and variances from phases 5 and 2.

| Hurst Exponent | Elevation | Slope | Var_Ph5 | Var_Ph2 |
|:---:|:---:|:---:|:---:|:---:|
| H2 | −0.81 | −0.53 | 0.54 | 0.29 |
| HI | −0.25 | −0.07 | 0.05 | 0.21 |

When H2 was used, the three-clusters analyses presented more significant differences. These differences are shown in their dynamics, as seen in the variances calculated separately for each cluster, phase, and NDVI (Figure 10), where some pixels were distinct. These differences were still found when all pixels were averaged for each cluster (Figures 8 and 10). These differences in NDVI are reflected in the type of vegetation found dominating each pixel. Cluster 1, where we found the majority of pixels, reflects a great variation from woodlands to grasslands. In this region with an arid climate, patchy landscapes with different vegetation are typical and they can occur along an ecological continuum, rather than as well-defined and separated ecosystems [76,77]. Cluster 2 shows a vast majority of woodland, while cluster 3 consists mainly of grassland (Table 3), despite cluster 1 having both grassland and woodland, as reflected by an intermediate average NDVI for cluster 1. Pixels from cluster 2 are those with higher NDVI representing thicker forests, unlike the more dispersed forests with shrubs found in cluster 1.

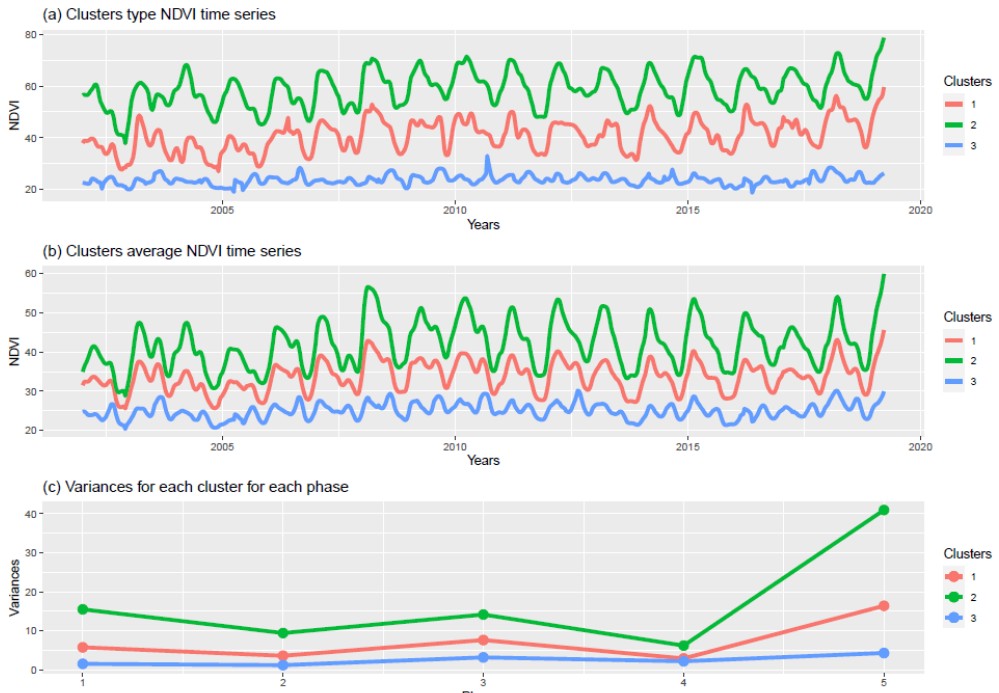

**Figure 10.** Time series of cluster prototypical pixels in: (**a**) the cluster type (selected based on their vegetation type: mixed shrubland for cluster 1, open woodland for cluster 2 and grassland for cluster 3); (**b**) the average of each cluster; and (**c**) the variances for each cluster and phase, based on the NDVI dynamics following [64].

**Table 3.** Percentages of vegetation type of the selected pixels, based on the National Forest Map. Results for each cluster are based on URF with H2.

| Cluster | Woodland | Shrubland | Grassland |
|---|---|---|---|
| 1 | 48.0 | 7.7 | 44.3 |
| 2 | 99.7 | 0.3 | 0.0 |
| 3 | 15.5 | 4.4 | 80.1 |

The Mann–Kendall test was performed for all the pixels and the area. Although all three clusters showed that most pixels had a significant positive trend, cluster 2 had 90% of the pixels in that category, while clusters 1 and 3 only had 67% and 64%, respectively (Table 4).

**Table 4.** Percentages of Mann–Kendall results for each cluster based on URF with H2.

| Significance | Cluster 1 | Cluster 2 | Cluster 3 |
|---|---|---|---|
| Significant decrease | 6.2% | 0 % | 2.2% |
| Not significant | 26.5% | 10% | 34.3% |
| Significant increase | 67.3% | 90% | 63.5% |

## 4. Discussion

The link between elevation and the Hurst exponent was previously reviewed by Peng [5], who found a good relationship between HI and elevation. In our study, the stronger correlation of H2 with NDVI time-series variances, compared with HI, suggests the importance of detrending in fractal analyses when studying vegetation time series. Differences between R/S and DFA were previously reported [50], as DFA is less affected by size effects or spurious correlation of non-stationary time series [50,78]. Our results support these findings, highlighting the relevance of detrending, especially when studying different vegetation types. Limited differences in pixel clustering were found in both methods of calculating the Hurst exponent in areas dominated by grasslands, suggesting that a tendency is not present in this NDVI series probably due to the grazing effect on these areas. On the other hand, more significant differences in areas with more trees were found. In this case, grazing does not limit the vegetation growth of trees, showing a trend in their vegetation time series.

Arid rangelands are spatially heterogeneous [4,26], and land degradation and over-grazing can affect the landscape creating a grassland/woodland continuum [79,80]. This effect is reflected in the overlapping clusters, showing that discrete areas can have similar vegetation. However, differences among the majority of the pixels of each cluster in persistence, elevation, and slope were found. In further research, other factors relating to elevation and slope could be considered, such as availability for machine use in agriculture (easier on flatter areas), rainfall, soil depth, or erosion. These factors should be considered in land management.

Clustering vegetation dynamics and comparing those clusters with vegetation type illustrate the tendencies related to each vegetation. Understanding these processes is key to the spatiotemporal interactions between human and natural systems [18,19]. Most pixels were categorised as antipersistent and with a significantly increasing trend. Land managers should make special efforts to avoid further land degradation. Pixels categorised as the least antipersistent and with an increasing NDVI trend (as no persistent pixels were found) can be used as reference. These pixels can be studied to see if different management practices are in place leading to differences in persistence and NDVI trends.

The variability in arid areas was expected since minor changes in slope, rainfall, or other characteristics, mean a significant difference in water availability and plant growth [81,82]. Using URF to study rangelands can improve our understanding of the area even when fieldwork is unavailable, highlighting areas with different dynamics, crucial when monitoring vegetation. These techniques can also cluster a more extensive range of

land uses, not only be limited to rangeland, since they will have more distinctive spectral signatures. Further research should be made in other arid areas to contrast whether this method can allow us to analyse previous land classification, prioritise areas for future surveys, and improve management action.

This study includes several limitations. (1) MODIS spatial resolution is much larger than most land plots in this region. Despite remote-sensing data with a higher spatial resolution, 250 m spatial resolution was chosen, as it is used for indexed agricultural insurance in Spain. (2) Ground field visits were not possible to formally validate our results, and the Silhouette Index was used to compare the clustering results. Future steps could be to formally visit different areas for each cluster to validate these results. However, this study aids the body of research [5,6,8,44,45,50] supporting the use of persistence (DFA) and trends (Mann–Kendall) for vegetation series, using these techniques in arid rangelands to aid rangeland managers and policymakers.

## 5. Conclusions

Two methods (R/S and DFA) were used to calculate the Hurst exponent (HI and H2). The results were compared using two clustering methods, with summary statistics from the NDVI time series. The combination providing the best results was obtained based on the Silhouette Index and cluster characteristics. URF with the Hurst exponent from DFA (H2) showed the best outcome, compared with URF performed with the Hurst exponent calculated with R/S (HI), URF made without the Hurst exponent, and all the k-means results.

URF found differences when different Hurst exponent methods were used, while k-means found no differences. URF with H2 showed greater differences between areas with higher tree coverage and those with a mix of grassland and shrubland. Additionally, the H2 time series presented a stronger linear correlation with slope and elevation, an essential aspect of vegetation dynamics in arid environments.

Detrended fluctuation analyses produced significant differences when calculating the Hurst exponent in time series that presented a tendency. Detrending time series can allow for a better understanding of the dynamics of vegetation time series, as well as rangeland evolution and future trends. Rangeland persistence was a key aspect to consider in rangeland management and research. Thus, future research should explore more rangeland, and other land uses, and compare different land management practices.

**Author Contributions:** Conceptualization, A.M.T., A.S.-R., C.H.D.-A. and E.S.; methodology, A.M.T., A.S.-R., E.S. and E.S.; formal analysis, A.M.T., J.J.M.S., A.S.-R., M.R.-R. and E.S.; writing—original draft preparation, E.S.; writing—review and editing, E.S., J.J.M.S., A.S.-R., C.H.D.-A., M.R.-R., A.R. and A.M.T., visualization, A.M.T., A.R. and E.S.; supervision, A.M.T.; funding acquisition, A.M.T. All authors have read and agreed to the published version of the manuscript.

**Funding:** This work was partially funded by Boosting Agricultural Insurance based on Earth Observation data–BEACON project under agreement No. 821964, funded under H2020_EU, DT-SPACE-01-EO-2018-2020 and the Ministerio de Ciencia e Innovación (grant no. AGRISOST-CM S2018/BAA-4330). The authors also acknowledge support from Project No. PID2021-122711NB-C21 of the Ministerio de Ciencia, Innovación y Universidades of Spain.

**Institutional Review Board Statement:** Not applicable.

**Informed Consent Statement:** Not applicable.

**Acknowledgments:** The data provided by ENESA, the Ministerio de Agricultura, Pesca y Alimentación is greatly appreciated.

**Conflicts of Interest:** The authors declare no conflict of interest. The funders had no role in the design of the study; in the collection, analyses, or interpretation of data; in the writing of the manuscript, or in the decision to publish the results.

## Appendix A. Principal Component Analyses Made to Select the Most Explanatory Variables

**Table A1.** Three first principal components from the PCA for all the NDVI time-series variables and H2. Background color highlights those variables with higher explanatory power for each principal component.

| Variables | PC1 | PC2 | PC3 |
|---|---|---|---|
| NDVI_H2 | 0.14 | −0.27 | 0.84 |
| Var_Ph2 | 0.13 | −0.59 | −0.22 |
| Var_Ph3 | 0.13 | −0.56 | −0.37 |
| Var_Ph5 | 0.23 | −0.38 | 0.17 |
| Quartile_1_Ph2 | 0.32 | 0.01 | 0.03 |
| Quartile_1_Ph3 | 0.31 | 0.19 | −0.13 |
| Quartile_1_Ph5 | 0.31 | 0.16 | 001 |
| Median_Ph2 | 0.32 | 0.05 | 0.2 |
| Median_Ph3 | 0.31 | 0.14 | −0.14 |
| Median_Ph5 | 0.31 | 0.11 | 0.04 |
| Quartile_3_Ph2 | 0.32 | −0.01 | 0.01 |
| Quartile_3_Ph3 | 0.32 | 0.08 | −0.17 |
| Quartile_3_Ph5 | 0.33 | 0.05 | 0.07 |
| Standard deviation | 3.10 | 1.36 | 0.95 |
| Proportion of Variance | 0.74 | 0.14 | 0.07 |
| Cumulative Proportion | 0.74 | 0.88 | 0.94 |

**Table A2.** After removing the variables with a strong correlation for the summary statistics with the least explanatory power, the first principal components from the PCA for the selected variables are shown. Those variables with higher explanatory power for each principal component are highlighted.

| Variables | PC1 | PC2 | PC3 |
|---|---|---|---|
| NDVI_H2 | 0.36 | −0.54 | 0.74 |
| Var_Ph5 | 0.47 | 0.45 | 0.13 |
| Var_Ph3 | 0.43 | 0.54 | 0.07 |
| Var_Ph2 | 0.54 | −0.13 | −0.16 |
| Quartile_3_Ph5 | 0.12 | −0.43 | −0.64 |
| Standard deviation | 1.72 | 1.02 | 0.76 |
| Proportion of Variance | 0.59 | 0.21 | 0.12 |
| Cumulative Proportion | 0.59 | 0.79 | 0.91 |

## Appendix B. The Silhouette Indexes Calculated for All URF Changing Mtry and Number of Trees for Three Clusters

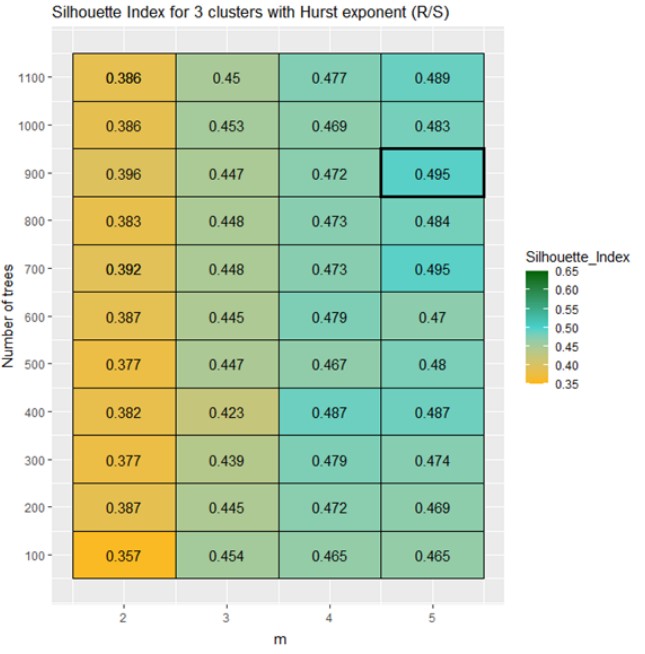

**Figure A1.** Silhouette indexes for three clusters for URF using Hurst exponent calculated with Rescaled Range method. "m" represents the number of predictors tested on each node.

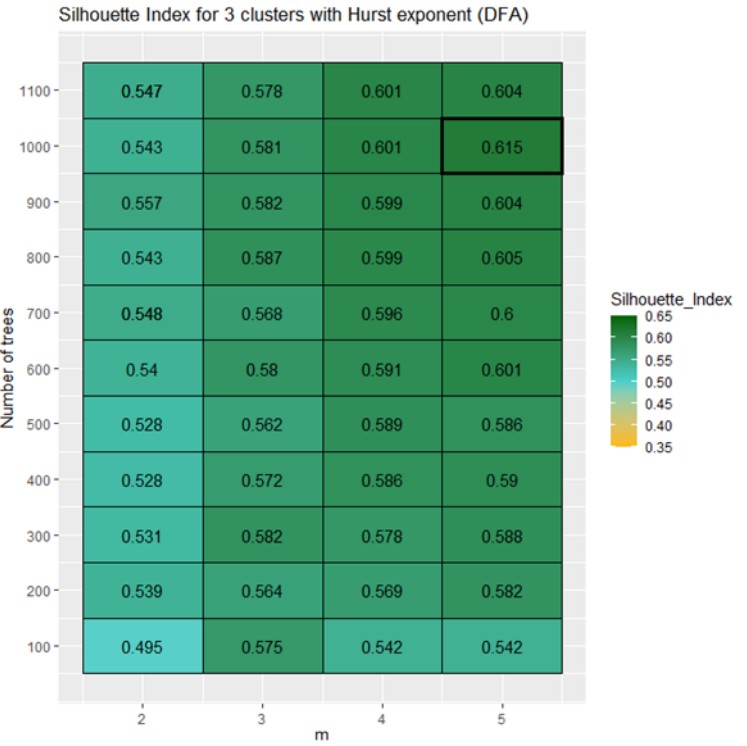

**Figure A2.** Silhouette indexes for three clusters for URF using Hurst exponent calculated with Detrended fluctuation analysis. "m" represents the number of predictors tested on each node.

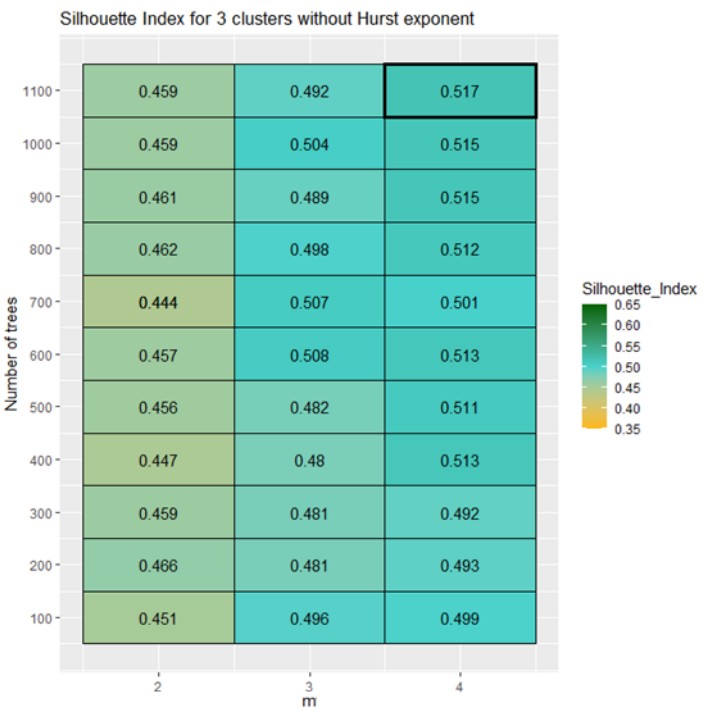

**Figure A3.** Silhouette indexes for three clusters for URF not using any Hurst exponent. "m" represents the number of predictors tested on each node.

**Appendix C. Maps of the Clusters Using URF with HI and H2 for the Three Study Provinces**

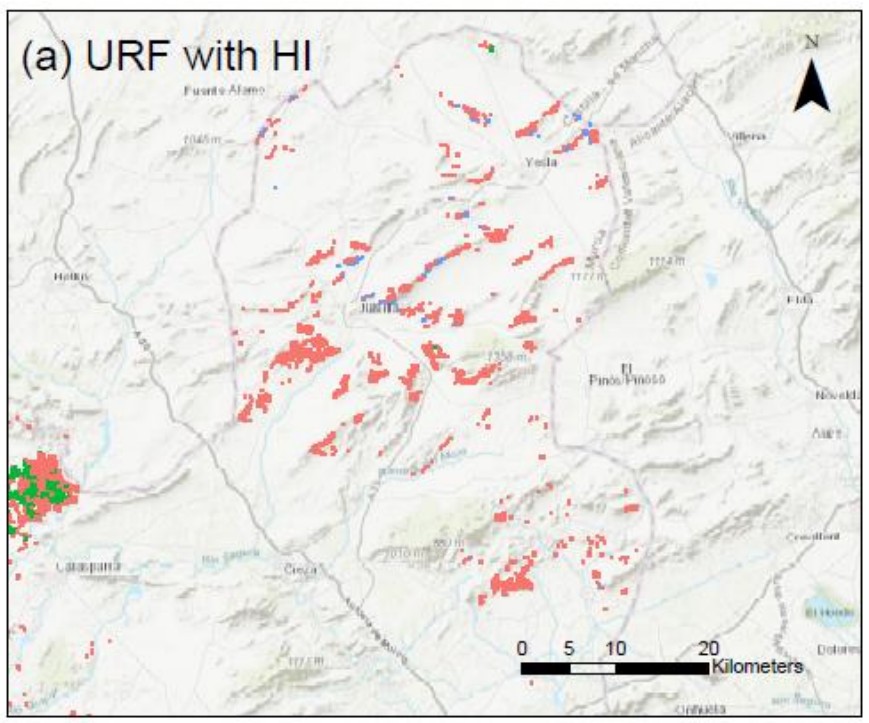

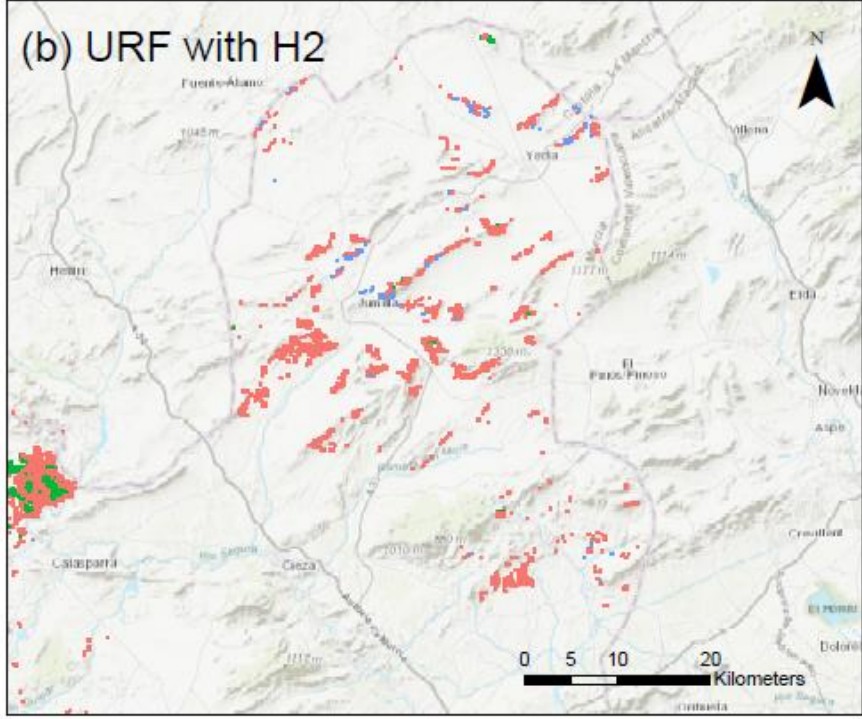

**Figure A4.** Comparison of the clustering results for URF using HI (**a**) and H2 (**b**) in the agricultural region of Murcia-NE. Cluster 1 is pink, cluster 2 is green, and cluster 3 is blue.

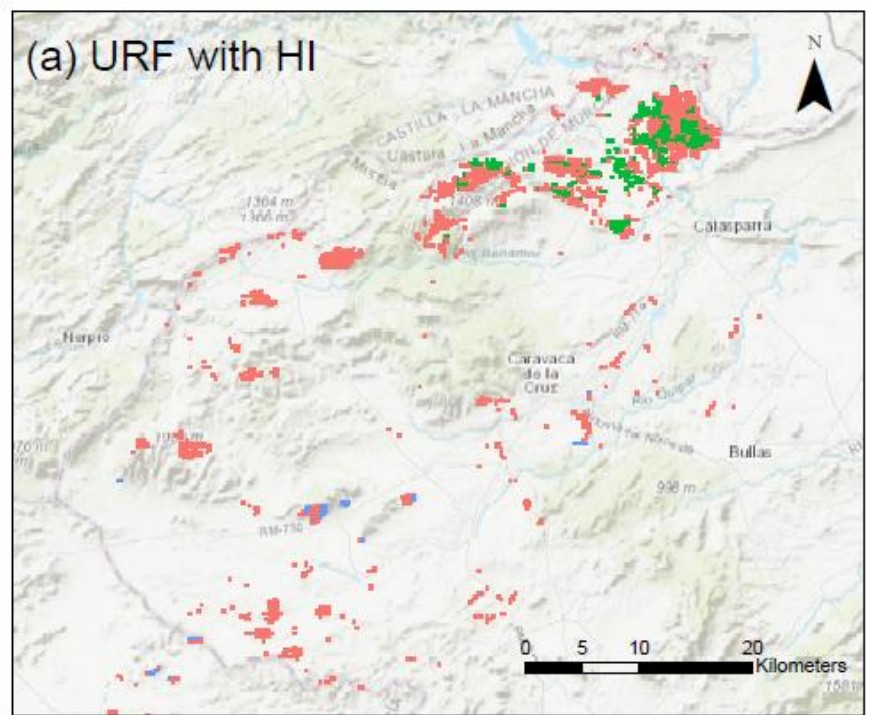

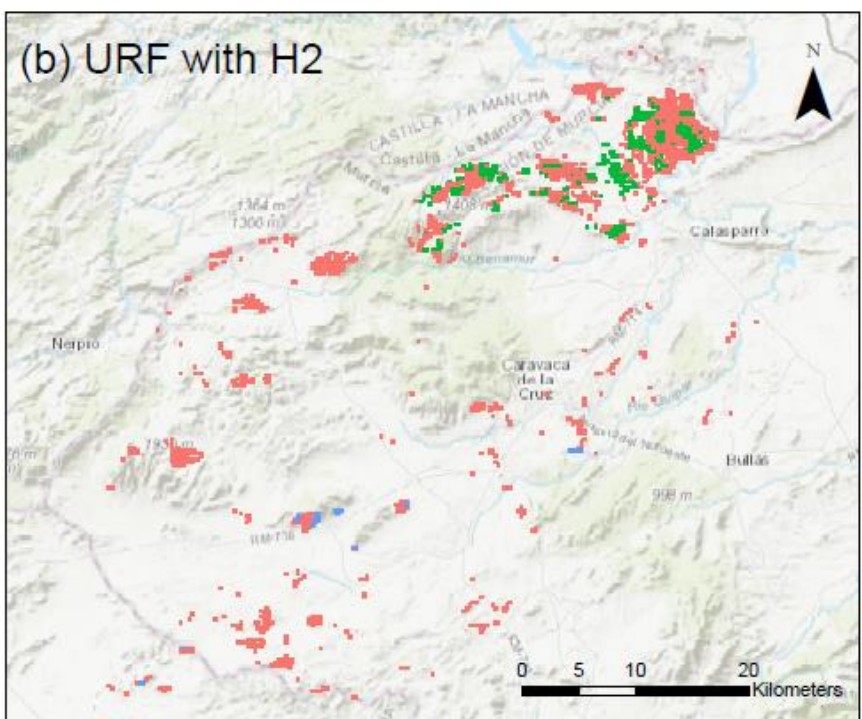

**Figure A5.** Comparison of the clustering results for URF using HI (**a**) and H2 (**b**) in the agricultural region of Murcia-NW. Cluster 1 is pink, cluster 2 is green, and cluster 3 is blue.

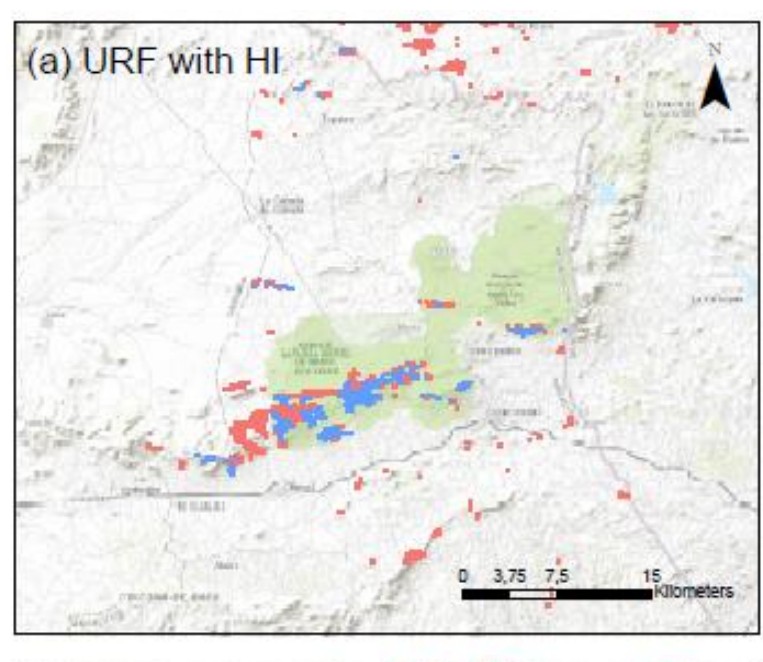

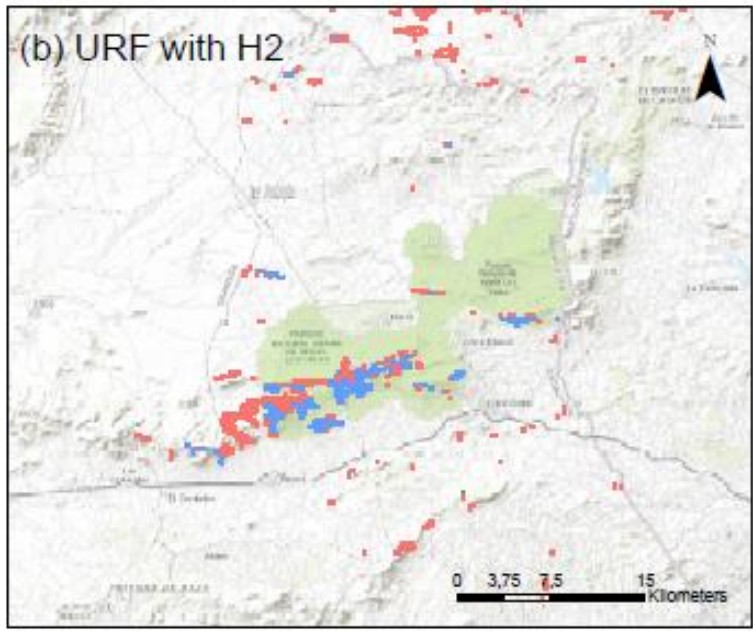

**Figure A6.** Comparison of the clustering results for URF using HI (**a**) and H2 (**b**) in the agricultural region of Los Vélez (Almería). Cluster 1 is pink, cluster 2 is green, and cluster 3 is blue.

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
