# Peer review of "Clustering Arid Rangelands Based on NDVI Annual Patterns and Their Persistence"

_remotesensing, doi:10.3390/rs14194949_

Round 1

Reviewer 1 Report

Using MODIS NDVI products to classify the rangelands by NDVI dynamics based on HURST exponent calculation has been proved to be feasible and effective. The manuscript is generally well written, but I have three concerns:1) What is the novelty of this research? 2) The spatial resolution of the data used is low, why the authors don’t use more fine resolution data? Such as Landsat and Sentinel-2?  3) Why using topographic data to compare the results?

Detailed comments:

1.         The title is not appropriate and the meaning is not clear

2.         Line 31-32:Grammatical mistake, unclear meaning.

3.         Introduction: the key scientific questions are not addressed directly.

4.         Line 96-97:Confused descriptions.

5.         Figure 1: the legend is missing.

6.         Dataset: NDVI product with 250m spatial resolution seems to be too course, and there are only 3654 pixels in the three areas. For rangeland dynamics, the patches could be dispersed distributions and the significant changes of NDVI is probably in the small plots.

7.         Line 133-134: It’s better to replace’ a 250 x 250 m2 spatial resolution’ by ’a 250m spatial resolution’.

8.         Line 126-129; what is the resolution of Dem data? And what is ‘those used in the clustering analysis’?

9.         Formula 2: what does the parameter ‘c’ express?

10.     Figure 4 and figure 6: what does the x axis present?

11.     Methods: Why use elevation and slope to compare the clustering results?

12.     Line 354-355: what does it mean?

13.     Line 364-365: the vegetation types and coverage information should be given to support this statement.

Reviewer 2 Report

The text seems interesting in the meaning of methods' orchestration. It is well-structured and clear for understanding.

Minor spell check is needed. Particularly, in row 42 - "recently research" is it ok, or "recent research"/"recent studies" phrase can be used?

Two citing styles are used in the text in parallel "Name, year" and "[number in the references list]" while stylistically it is better to use one.

Reviewer 3 Report

- The study focused on arid rangelands classification using Hurst Exponent for understanding vegetation dynamics.

- The study represents a good contribution to the field of applied remote sensing for monitoring of vegetation dynamics, and applying machine earning methods.

- Why did you consider only NDVI index? what about EVI and other sensitive indicators of vegetation change in arid and semi-arid environments?

- Why did you select unsupervised random forest and not supervised random forest in your classification?

- it is recommended to include flowchart to show the methodology adopted in this study, as well as provide more details regarding the processing of the remote sensing data?

- Why did you select the time period from 2000 to 2019 as your study period? is it because of the availability of MODIS data during this time period? did you realized unusual pattern of change might be because of drought event or any other muman didturbance to be compared with a normal conditions?

- Did you perform field visits to the study area to validate the remote sensing data analysis? as well as identifying the different classes in the study area for classification? if yes, then provide more details with field photographs.

Round 2

Reviewer 1 Report

Although all the comments were addressed in the revised version, I still find some wrong or inappropriate expressions as follows:

1.         Introduction: what is the main contributions of this research to scientific problems or application value? The revised version just gave an explanation of the methodology used.

2.         Line 371-377: grammar mistakes.

3.         Line 423-425: unclear statements.

        4.          Line 431-432: confused statements.

Reviewer 3 Report

 Accept in present form.

Author Response

Thank you very much for your time and attention.
